bioengineering

quadruped running, interlimb coordination, Central pattern generator, local sensory feedback

**Author for correspondence:**
Akira Fukuhara
e-mail: a.fukuhara@riec.tohoku.ac.jp

†Present address: Department of Mechanical Engineering, Osaka University, Suita, Japan.

Special Feature: Stability and manoeuvrability in animal movement: lessons from biology, modelling and robotics. Guest edited by Andrew Biewener, Richard Bomphrey, Monica Daley and Auke Ijspeert.

# Simple decentralized control mechanism that enables limb adjustment for adaptive quadruped running

Akira Fukuhara[1], Yukihiro Koizumi[1], Tomoyuki Baba[1], Shura Suzuki[1,2,†], Takeshi Kano[1] and Akio Ishiguro[1],

[1]Research Institute of Electrical Communication, Tohoku University, Sendai, Japan
[2]Japan Society for Promotion of Science, Tokyo, Japan

AF, 0000-0003-4303-7594; SS, 0000-0002-5392-2346; TK, 0000-0002-2033-4695; AI, 0000-0003-2850-0149

Quadrupeds exhibit versatile and adaptive running by exploiting the flying phase during the stride cycle. Various interlimb coordination mechanisms focusing on mechanical loads during the stance phase have been proposed to understand the underlying control mechanism, and various gait patterns have been reproduced. However, the essential control mechanism required to achieve both steady running patterns and non-steady behaviours, such as jumping and landing, remains unclear. Therefore, we focus on the vertical motions of the body parts and propose a new decentralized interlimb coordination mechanism. The simulation results demonstrate that the robot can generate efficient and various running patterns in response to the morphology of the body. Furthermore, the proposed model allows the robot to smoothly change its behaviour between steady running and non-steady landing depending on the situation. These results suggest that the steady and non-steady behaviours in quadruped adaptive running may share a common simple control mechanism based on the mechanical loads and vertical velocities of the body parts.

## 1. Introduction

Quadrupeds instantly change their locomotor patterns (e.g. gait patterns) to achieve an efficient and adaptive translation. For example, cursorial quadrupeds probably change their gait pattern from walking to trotting to galloping for sufficient energetics at various locomotion speeds [1,2]. In addition to steady locomotion, they can exhibit non-steady behaviour, such as leaping and landing, to overcome uneven terrain such as chasms and puddles. These adaptive locomotor behaviours are achieved by coordination between limbs, which is referred to as interlimb coordination. Decoding the interlimb coordination mechanism underlying a quadruped's versatile locomotion sheds new light on the development of control schemes for adaptive quadruped robots.

Neurophysiology and bioinspired studies regarding robotics suggest that decentralized coordination mechanisms partially generate interlimb coordination underlying the versatile locomotion of quadrupeds. Using decerebrate cats, Shik *et al.* demonstrated that a central pattern generator (CPG) and local reflexes allowed quadrupeds to exhibit gait transition in response to increments in locomotion speed [3]. Based on these biological findings [3,4], studies have elucidated the essential control mechanism by building mathematical and robotic models [5]. Consequently, studies have demonstrated that sensory feedback mechanisms based on ground reaction force (GRF) play essential roles in regulating versatile gait patterns for steady locomotion. However, the essential control mechanism involving non-steady locomotor behaviours, such as leaping and landing, remains unclear.

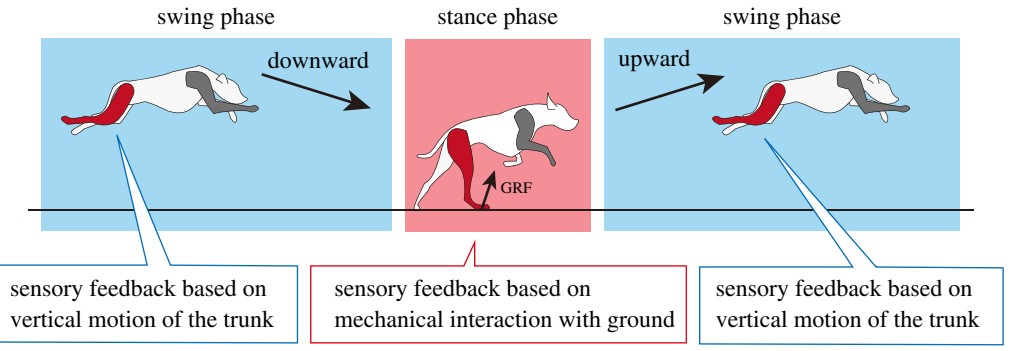

**Figure 1.** Concept of interlimb coordination mechanism with sensory feedback during stance and swing phases. During the stance phase, the motion of the limb is mainly modulated in response to the GRF to support the body weight. By contrast, during the swing phase, the motion of the limb is mainly modulated in response to the vertical motion to prepare for the next touchdown event. (Online version in colour.)

The behavioural observation of a running quadruped indicates that it is possible to modulate locomotor patterns during the stance and swing phases for steady and non-steady locomotion, respectively. During gait transition and overcoming objects [6–8], quadrupeds probably regulate the phase relationship between limbs by adjusting limb touch-down timing in the swing phase. A few studies have considered the postural reflexes for bioinspired interlimb coordination to realize posture stabilization in rough terrain [9–11] and gait transition [12,13]. However, the essential control mechanism required during the swing phase for steady and non-steady quadruped locomotion involving leaping and landing remains unclear.

To address the aforementioned problem, we reconsider our simple decentralized interlimb coordination mechanism involving modulation in the stance and swing phases. Despite the absence of neural coupling between limbs, our previous CPG models demonstrated speed-dependent gait transition using a simple local sensory feedback mechanism [14,15]. Consequently, it has been elucidated that physical communication between limbs through GRF is essential for phase modulation during stance. Based on this minimal model approach, this study proposes a new interlimb coordination mechanism focusing on sensory feedback during the stance and swing phases for steady and non-steady quadruped locomotion, respectively. Specifically, our new model exploits sensory information regarding the GRF and the velocities of each body part to realize faster-running patterns and adaptability to falling to the ground during a bounding gait. These results suggest that the vertical velocities of each body part are essential for changing the interlimb coordination from the periodic behaviour for running and preliminary behaviour for landing from an elevated location.

## 2. Methods

### (a) Concept

Sensory-motor systems realize adaptive quadruped running over various layers, such as several sensory organs, the higher nervous system (e.g. brain), distributed nervous system (e.g. CPG local reflexes) and intelligent mechanical system. For example, local reflexes through the spinal cord and the compliant limb structure allow animals to adapt to perturbations from the ground, such as collisions with obstacles in the swing phase [16]. In addition, the closed-loop sensory-motor system,

including the brain stem, exploits various sensory information, such as the vestibular system, visual information, and somatic sensation, to modulate motor commands for the stabilization of body posture and gaze [17–19]. Although each closed-loop mechanism appears to be significant in quadruped locomotion, the integration of these complex mechanisms to achieve fast and adaptive quadruped running remains unclear.

To extract the mechanism underlying adaptive quadruped running, we focus on the physical phenomena in quadruped running, rather than the biological details, and construct a simple interlimb coordination mechanism. Each limb exhibits a leaping motion during fast running: the limb kicks the ground in the stance phase, the body moves upward and then changes its motion downward due to gravity, which is followed by ground contact (figure 1). Especially in the swing phase, the adjustment of the limb's posture and phase of the stride motion is critical to prepare the initial conditions for the subsequent stance phase. Based on the leaping motion of each limb, we hypothesize that the vertical movement of the body (e.g. moving upward or downward) contains rich information to modulate limb motion. Based on this hypothesis, we developed an interlimb coordination mechanism in a decentralized manner similar to our previous study [14,15]; thus, there is no neural connectivity between the limbs, and each limb generates its motion according to the descending commands and local sensory feedback mechanisms. In the remainder of this section, we first explain a simple robot structure, basic limb motion, and the interaction between the body and environment. We then model the decentralized interlimb coordination mechanism.

### (b) Robot model
#### (i) Overview

To build a minimal interlimb coordination model, the mechanical system is simplified as the sagittal-plane model shown in figure 2a. The body consists of two limbs (i.e. fore and hind limbs) and a trunk with a single pitch joint, and it is described by a mass–spring–damper system where point masses are located at the shoulder, spine, hip, fore foot and hind foot. The point masses are connected to a spring and damper. Each limb has two degrees of freedom. Specifically, the shoulder (hip) joint consists of a parallel combination of a rotary spring and damper, while the limb expands and contracts by a parallel combination of a prismatic spring and damper. Considering the mass distribution of real quadrupeds, the weight of the foot point mass is set to be smaller than those of the point masses in the trunk of the body. Furthermore, the body trunk has three point masses ($m_{\text{fore}}^{\text{base}}$, $m^{\text{spine}}$ and $m_{\text{hind}}^{\text{base}}$), to adjust the mass distribution. These point masses in the trunk are connected by passive springs and dampers. While some mammal species (e.g. cheetahs)

**Figure 2.** Mechanical model of a 2D robot. (*a*) Mass–spring–damper system. Each point mass is located at each joint in the fore and hind feet. The weight of each foot point mass is set to a light value (e.g. foot point mass is 0.6 kg, while each body point mass is 17 kg). The foot point mass is actuated by a prismatic and rotary spring and damper, while the links in the body remain rigid with stiff springs and dampers, (*b*) foot trajectory and (*c*) phase oscillator for limb control. The reference position for the foot is determined by the phase oscillator. The ground reaction force vector $\mathbf{N}_i$ applied at the *i*th limb consists of vertical and horizontal components ($N_i^v$ and $N_i^h$). (Online version in colour.)

exploit the flexibility of the trunk, we employ a rigid trunk for simplicity. The coefficients of the prismatic spring and damper that connect the spine point mass and shoulder (hip) are set to be large so that the link of the trunk remains rigid as follows:

$$\tau_{\text{trunk}} = K_{\text{trunk}}^{\text{rot}}(\bar{\theta}_{\text{trunk}} - \theta_{\text{trunk}}) - D_{\text{trunk}}^{\text{rot}} \dot{\theta}_{\text{trunk}} \tag{2.1}$$

and

$$F_{\text{trunk},i} = K_{\text{trunk}}^{\text{pri}}(\bar{L}_{\text{trunk},i} - L_{\text{trunk},i}) - D_{\text{trunk}}^{\text{pri}} \dot{L}_{\text{trunk},i}, \tag{2.2}$$

where $\tau_{\text{trunk}}$ and $F_{\text{trunk},i}$ are the applied torque and force at the rotary and prismatic springs in the middle of the trunk, respectively, $K_{\text{trunk}}^{\text{rot}}$ and $D_{\text{trunk}}^{\text{rot}}$ are the spring and damper coefficients around the spine point mass, respectively, $K_{\text{trunk}}^{\text{pri}}$ and $D_{\text{trunk}}^{\text{pri}}$ are the spring and damper coefficients for the trunk links, respectively, $\bar{\theta}_{\text{trunk}}$ and $\bar{L}_{\text{trunk},i}$ are the natural angle and length of the rotational spine joint and body link, respectively. The index *i* represents the fore and hind limbs (i.e. *i* = fore, hind).

### (ii) Foot trajectory

Each leg is actuated by changing the target angle of the rotary spring at the shoulder (hip) joint and the target length of the prismatic springs for the limbs. Although the foot trajectories of real cursorial mammals change depending on the situation owing to intralimb coordination mechanisms [20], here we assume that the foot traces a simple specific trajectory, as shown in figure 2*b*, to concentrate on the interlimb coordination mechanism. A phase oscillator is employed to describe the periodic motion of the limb. In particular, the position of the foot is described using the oscillator phase $\phi_i$ (*i* = fore, hind) as follows:

$$\bar{\theta}_i = \theta_i^{\text{offset}} + C^{\text{rot}} \cos \phi_i \tag{2.3}$$

and

$$\bar{L}_i = \begin{cases} L^{\text{offset}} - C_{\text{sw}}^{\text{pri}} \sin \phi_i & (\sin \phi_i > 0), \\ L^{\text{offset}} - C_{\text{st}}^{\text{pri}} \sin \phi_i & (\text{otherwise}), \end{cases} \tag{2.4}$$

where *i* denotes the index of the limbs, $\bar{\theta}_i$ is the target angle of the shoulder (hip) joint, $\theta_i^{\text{offset}}$ is a constant angular offset of the limb joint, $C^{\text{rot}}$ is a constant that defines the amplitude of rotary motion, $\bar{L}_i$ is the target length of the *i*th limb, $L^{\text{offset}}$ is a constant defining the offset of the prismatic joint, and $C_{\text{st}}^{\text{pri}}$ and $C_{\text{sw}}^{\text{pri}}$ denote constants that determine the amplitude of prismatic motion during expansion and contraction, respectively. When $\sin \phi_i > 0$, the limb tends to contract and lift off the ground (i.e. the swing phase in figure 2*c*). Conversely, when $\sin \phi_i < 0$, the limb tends to expand and remain on the ground to support the body (i.e. the stance phase in figure 2*c*). Here, $C_{\text{sw}}^{\text{pri}}$ is designed to be larger than $C_{\text{st}}^{\text{pri}}$ to ensure ground clearance during the swing phase.

The torque generated at the shoulder (hip) joint $\tau_i$ and the force generated by the parallel combination of the prismatic

spring and damper $F_i$ are calculated as follows:

$$\tau_i = K_{\text{limb}}^{\text{rot}}(\bar{\theta}_i - \theta_i) - D_{\text{limb}}^{\text{rot}} \dot{\theta}_i \tag{2.5}$$

and

$$F_i = K_{\text{limb}}^{\text{pri}}(\bar{L}_i - L_i) - D_{\text{limb}}^{\text{pri}} \dot{L}_i, \tag{2.6}$$

where $K_{\text{limb}}^{\text{rot}}$ and $D_{\text{limb}}^{\text{rot}}$ are the spring and damper coefficients of the hip (shoulder) joint of the limb, respectively, $K^{\text{pri}}$ and $D^{\text{pri}}$ are the spring and damper coefficients of the prismatic joint of the limb, respectively, $\theta_i$ is the actual angle of the hip (shoulder) joint, and $l_i$ is the actual length of the limb.

### (iii) Interaction between the leg tip and the ground

The interaction between the leg tip and the ground is complex because both static and dynamic friction must be considered. However, for simplicity, we neglect the static friction and describe the physical interaction between the foot point mass and the ground as follows:

$$N_i^v = \begin{cases} -K^{\text{gnd}} y_i - D^{\text{gnd}} \dot{y}_i^{\text{foot}} & (y_i^{\text{foot}} < 0), \\ 0 & (\text{otherwise}) \end{cases} \tag{2.7}$$

and

$$N_i^h = \mu N_i^v(-\tanh \eta \dot{x}_i^{\text{foot}}), \tag{2.8}$$

where $N_i^v$ and $N_i^h$ are the vertical and horizontal components of the GRF applied at the *i*th limb, respectively, $K^{\text{gnd}}$, $D^{\text{gnd}}$ and $\eta$ are positive constants, $\mu$ is the friction coefficient, and $y_i^{\text{foot}}$ and $x_i^{\text{foot}}$ denote the vertical and horizontal positions of the *i*th foot point mass, respectively. Note that equation (2.8) describes the Coulomb friction when $|\dot{x}_i^{\text{foot}}| \gg \eta^{-1}$.

## (c) Decentralized control mechanism for quadruped running

This study attempts to generate flexible interlimb coordination patterns using a decentralized control mechanism that exploits the body dynamics of the quadruped robot model while running. The decentralized coordination mechanism was designed based on the model proposed by Owaki & Ishiguro [14]. In their model, the oscillator phase is modified such that each leg remains in the stance phase when it detects a GRF. However, because sensory feedback is applied only during the stance phase, the phase is not modulated during the swing phase. Here, we hypothesize that phase modulation during the swing phase is also important for increasing the stability of locomotion on an irregular terrain and introduce an additional sensory feedback term to Owaki & Ishiguro's model to describe phase modulation during the swing phase.

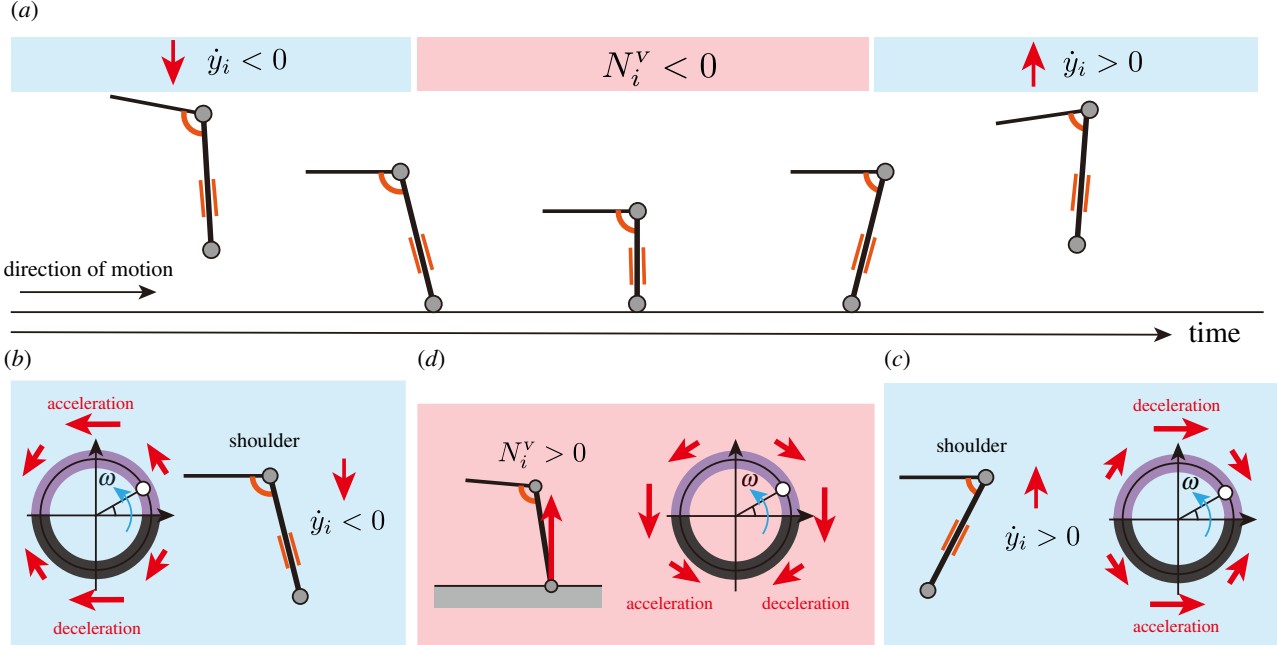

**Figure 3.** Sensory feedback mechanism for quadruped running. (*a*) The running sequence of the limb is moving downward in the swing phase, the stance phase, and moving upward in the swing phase. Sensory feedback based on (*b*) downward and (*c*) upward motion of the base of the limb during the stride cycle. (*d*) Sensory feedback based on GRF. (Online version in colour.)

From a biological perspective, the sensory information used for phase modulation during the swing phase remains unclear. Thus, we attempt to understand the significant control mechanism by describing how sensory feedback could work in a simple and abstract manner. Therefore, we assume that the information regarding the velocity of the shoulder (hip) point mass along the vertical direction, $\dot{y}_i$, is used to modulate the oscillator phase during the swing phase, although there is no proof that real quadrupeds have sensors detecting the velocity at the shoulder (hip).

Based on the aforementioned, we formulate a decentralized control mechanism with sensory feedback mechanisms. The time evolution of $\phi_i$ is described as follows:

$$\dot{\phi}_i = \omega - \sigma^{\mathrm{GRF}} N_i \cos \phi_i - \sigma^{\dot{y}} \dot{y}_i \sin \phi_i, \tag{2.9}$$

where $\omega$ denotes the intrinsic angular velocity, and $\sigma^{\mathrm{GRF}}$ and $\sigma^{\dot{y}}$ denote the gains for the GRF and velocity feedback, respectively. The first term on the right side of equation (2.9), namely $\omega$, corresponds to the descending command from the brain to the CPG network to regulate the locomotor frequency: a low $\omega$ value generates a slow limb stride motion, whereas a high $\omega$ value generates a fast limb stride motion.

The second term on the right side of equation (2.9) is a sensory feedback term based on the GRF applied at the $i$th limb, which was proposed by Owaki & Ishiguro [14]. This feedback term modulates the motion of each limb depending on the magnitude of the GRF, such that the loaded limb tends to continue supporting the body weight. Specifically, while $N_i > 0$, the second term on the right side of equation (2.9) causes $\phi_i$ to increase during the second and third quadrants, and $\phi_i$ decreases during the first and fourth quadrants.

The third term on the right side of equation (2.9) is proposed in this study. When $\dot{y}_i > 0$, the third term on the right side of equation (2.9) delays $\phi_i$ during the first and second quadrants, and $\phi_i$ advances during the third and fourth quadrants. By contrast, when $\dot{y}_i < 0$, the third term on the right side of equation (2.9) advances $\phi_i$ during the first and second quadrants, and $\phi_i$ is delayed during the third and fourth quadrants. Thus, this feedback term is expected to work as follows (figure 3). At an early stage of the swing phase, the phase velocity slows down and tends to remain in the swing phase because the body is expected to move up, that is, $\dot{y}_i > 0$. Meanwhile, at the late stage of the

swing phase, the body is expected to move down, that is, $\dot{y}_i < 0$; thus, the phase advances to $\pi$ to prepare for the next touchdown event. Owing to this feedback mechanism, the leg can reasonably adjust its rhythm during the swing phase.

## (d) Evaluation
### (i) Cost of transport
To evaluate the locomotor performance, we measured the locomotion speed and cost of transport (COT). The criterion, COT, defined by [21], is calculated as follows:

$$\mathrm{COT} = \frac{1}{Xmg} \int_0^T P(t)\, \mathrm{d}t, \tag{2.10}$$

where $X$ (m) is the distance travelled over a period $T$ (s), $m$ (kg) is the total mass of the robot, and $g$ (m s$^{-2}$) is the gravitational acceleration. The power consumption of the actuator $P$ [W] is estimated by referring to Nishii *et al.* [22] as follows:

$$P(t) = \sum_i \left( \chi(\tau_i(t)\dot{\theta}_i(t)) + \gamma^{\mathrm{rot}}\tau_i^2(t) + \chi(F_i(t)\dot{L}_i(t)) + \gamma^{\mathrm{pri}}F_i^2(t) \right) \tag{2.11}$$

and

$$\chi(z) = \begin{cases} 0 & (z \leq 0), \\ z & (z > 0), \end{cases} \tag{2.12}$$

where $\gamma$ is a positive constant related to the energy consumption due to heat emission, and the function $\chi(z)$ returns the value of argument $z$ if $z$ has positive values. In this simulation, the constant values for the rotary and prismatic actuators, $\gamma^{\mathrm{rot}} = 0.001$ and $\gamma^{\mathrm{pri}} = 0.01$, are determined such that the positive work at the actuator becomes nearly of the same order as that of the heat dissipation.

### (ii) Gait patterns
In this study, the emerging interlimb coordination patterns are evaluated based on the phase differences between the limbs and the number of distinct flying phases. Actual quadrupeds exhibit various phase differences between the four limbs depending on the gait patterns, for example, cantering, galloping, bounding and pronking for high-speed running [1]. In these running gaits, interlimb coordination patterns can be distinguished based on phase differences between the forelimb and hindlimb. For example,

in the pronking gait, the forelimbs and hindlimbs move in phase. In horse-like galloping, quadrupeds contact the ground with their hindlimbs first, and then take off the ground with their forelimbs [23,24]. Consequently, forelimb contacts have a short lag after the hindlimb contacts. By contrast, quadrupeds in cheetah-like galloping have a longer phase lag between the hindlimb contact and forelimb contact [23,24]. Based on these phase relationships between the forelimbs and hindlimbs, we evaluated the two-limbed quadruped robot system by measuring the phase difference between the forelimbs and hindlimbs, $\Delta\phi$. The parameter $\Delta\phi$ is evaluated using directional statistics, as follows:

$$\Delta\phi = \arg\left(\sum_{t=n_{\text{start}}}^{n_{\text{end}}} e^{j(\phi_{\text{hind}}(t)-\phi_{\text{fore}}(t))}\right), \tag{2.13}$$

where $n_{\text{start}}$ and $n_{\text{end}}$ are the time steps at the beginning and end of the measurement period, respectively. When $\Delta\phi$ is close to 0 or $2\pi$, the forelimbs and hindlimbs move synchronously, similar to the pronking gait. When $\Delta\phi = \pi/2$, there is a short phase lag between the forelimbs and hindlimbs (i.e. horse-like running). Furthermore, when $\Delta\phi = \pi$, there is a long phase lag between the forelimbs and hindlimbs (i.e. cheetah-like running).

In addition, we determined the number of flying phases (no ground contact) during one stride cycle. For example, the pronking gait has one major flying phase after synchronous stances by the forelimbs and hindlimbs. In horse-like running, there is one major flying phase after the forelimb stance. By contrast, cheetah-like running has two major flying phases after each forelimb and hindlimb stance.

## 3. Results

Three types of experiments were conducted to evaluate the proposed model. The first experiment was a running task with different combinations of feedback gains to address the locomotor performance (e.g. locomotion speed) via the proposed decentralized control mechanism. The second experiment involved a falling task during running to address real-time adaptation to changes in the environment. The third experiment was an adaptation to different morphologies to address the variation in the running patterns via the proposed model.

In all types of experiments, the size of the robot was that of a large dog; for example, the total body weight was approximately 50 kg, the height of the shoulder was 0.5 m, and the length of the trunk was 1.0 m. The other parameters, such as the coefficients of the spring and dampers for the body and ground, were set heuristically to allow the robot to run with a bounding gait. Consequently, the parameters were set as listed in electronic supplementary material, table S1.

As the initial condition of all experiments, the initial robot position was at a height of 0.5 m to prevent the robot from having any ground contact. Furthermore, the robot had no initial velocity of the body moving forward; therefore, the robot fell before the first touchdown event. Regarding the phase of the limb controller, the robot had a specific initial phase $(\phi_{\text{fore}}, \phi_{\text{hind}}) = (0.5\pi, 1.5\pi)$. The value of $\omega$ increased from 0.0 rad s$^{-1}$ to a specific value (e.g. 13.75 rad s$^{-1}$ in electronic supplementary material, table S1) at the beginning of the running simulation.

### (a) Emergence of efficient bounding gait

The first simulation experiment involved a running task. For comparison, we changed the feedback gains from $(\sigma^{\text{GRF}}, \sigma^{\dot{y}}) = (0.025, 0.0)$ to $(0.025, 5.0)$ and measured the locomotor speed and COT. The results in figure 4 demonstrate that

the sensory feedback in both the stance and swing phases allows the robot to generate faster and more efficient running patterns than those with the feedback in only the stance phase. Figure 4a shows the changes in the locomotion speed during a trial of the running task with the initial phase $((\phi_{\text{fore}}, \phi_{\text{hind}}) = (0.5\pi, 1.5\pi))$. While the intrinsic angular velocity $\omega$ is constant at 13.75 (rad s$^{-1}$), the robot increases the locomotion speed from 1.6 to 1.9 (m s$^{-1}$) owing to the effects of the sensory feedback mechanism based on the velocity of the body parts. Regarding efficiency, the robot improves the COT from 0.88 to 0.84. In ten trials with random initial phase values, the average locomotion speed increases from 1.57 to 1.96 (m s$^{-1}$) and the average COT value changes from 1.01 to 0.80, depending on the aforementioned changes in $\sigma^{\dot{y}}$ values.

The improvement in the locomotor performance is due to the change in the interlimb coordination between the fore and hind limbs. According to figure 4b, the robot changes the average phase difference between the limbs $\Delta\phi$ from $0.6\pi$ to $0.8\pi$ (rad). When $\Delta\phi$ is nearly $0.5\pi$, the forelimb touches the ground after the touchdown event of the hindlimb. While running with a sensory feedback based only on the GRF, the robot exhibits a phase difference between the fore and hind limbs $\Delta\phi$ of nearly $0.5\pi$. In this coordination pattern, the forelimb touches the ground after the touchdown event of the hindlimb, as shown in figure 4c. By contrast, the robot with a feedback based on both the GRF and vertical motion of the body exhibits a nearly antiphase coordination between the fore and hind limbs (e.g. $\Delta\phi = 0.8\pi$), as shown in figure 4d. In this coordination pattern, there are two flying phases after the stance phases of both the fore and hind limbs. Therefore, the velocity sensory feedback mechanism allows the robot to exploit the flying phase to increase the stride length.

Furthermore, we investigate the parameter dependencies on the gains of the sensory feedback terms $\sigma^{\text{GRF}}$ and $\sigma^{\dot{y}}$. Figure 5 presents the results of running with various combinations of feedback gains. The results indicate that faster and more efficient running is achieved using sensory feedback terms based on both the GRF and vertical motion of the body (e.g. $\sigma^{\text{GRF}} > 0$ and $\sigma^{\dot{y}} > 0$) than that using feedback terms based only on the GRF (e.g. $\sigma^{\text{GRF}} > 0$ and $\sigma^{\dot{y}} = 0$). Figure 4 also suggests that the two types of feedback terms should balance the values of their feedback gains to achieve fast and efficient running. For example, the combination of $\sigma^{\text{GRF}}$ of 0.025 and $\sigma^{\dot{y}}$ of 5.0 in figure 5 achieves faster and more efficient running than that with lower/higher values of feedback gains $\sigma^{\text{GRF}}$ and $\sigma^{\dot{y}}$. Regarding the phase difference between the fore and hind limbs, the robot with the sensory feedback based on the GRF exhibits a closed stride motion timing between them (e.g. $\Delta\phi = \pi/2$). By contrast, the sensory feedback based on the vertical motions of the body parts allows the robot to generate an antiphase coordination between the fore and hind limbs (e.g. $\Delta\phi = \pi$), and the combinations of the feedback gains allow faster and more efficient running motions. Consequently, the proposed sensory feedback mechanism regulates the interlimb coordination of the running robot for a faster and more efficient bounding gait.

### (b) Adaptation to changes in environment

According to a quadruped's versatile locomotion, the modulation of the swinging limb motion appears to contribute to the adjustment of the touchdown timing during the landing behaviour. Therefore, we conducted an experiment in which a running robot was made to fall to evaluate the effect of the

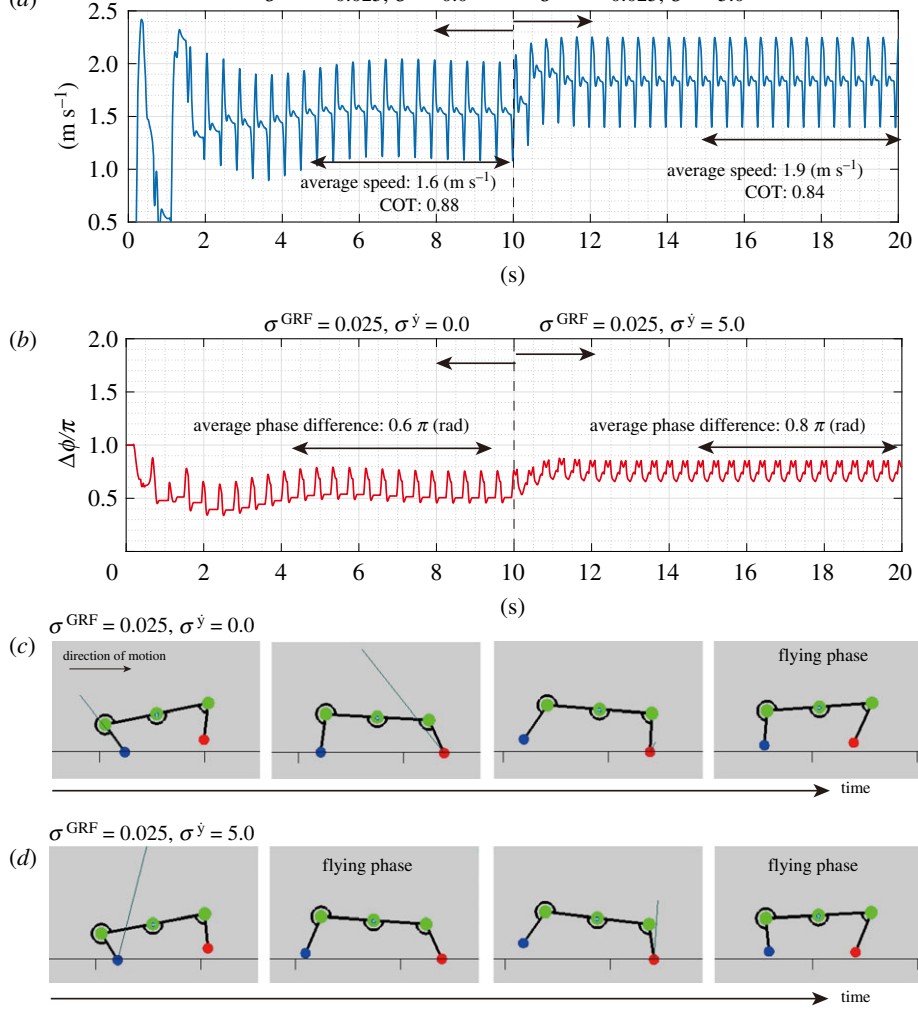

**Figure 4.** Comparison of the robot's running with and without the velocity feedback. This figure presents the result of a trial with an initial phase (($\phi_{\text{fore}}$, $\phi_{\text{hind}}$) = (0.5$\pi$, 1.5$\pi$)). (a) Locomotion speed. The robot runs with only the GRF sensory feedback (e.g. ($\sigma^{\text{GRF}}$, $\sigma^{\dot{y}}$) = (0.025, 0.0)) by 10 (s). Then, the robot runs with both the GRF and velocity sensory feedback (e.g. ($\sigma^{\text{GRF}}$, $\sigma^{\dot{y}}$) = (0.025, 5.0)), with the locomotion speed increasing from 1.6 (m s$^{-1}$) to 1.9 (m s$^{-1}$). (b) Phase difference between fore and hind limbs, $\Delta\phi = \phi_{\text{hind}} - \phi_{\text{fore}}$. The average phase difference between limbs, $\Delta\phi$, changes from 0.6$\pi$ to 0.9$\pi$ depending on the effects of the velocity sensory feedback. The intrinsic angular velocity $\omega$ is constant at 13.75 (rad s$^{-1}$) during the running experiment. The snapshots demonstrate the running patterns of the robot when $\Delta\phi = 0.6\pi$ (c) and 0.8$\pi$ (d). (Online version in colour.)

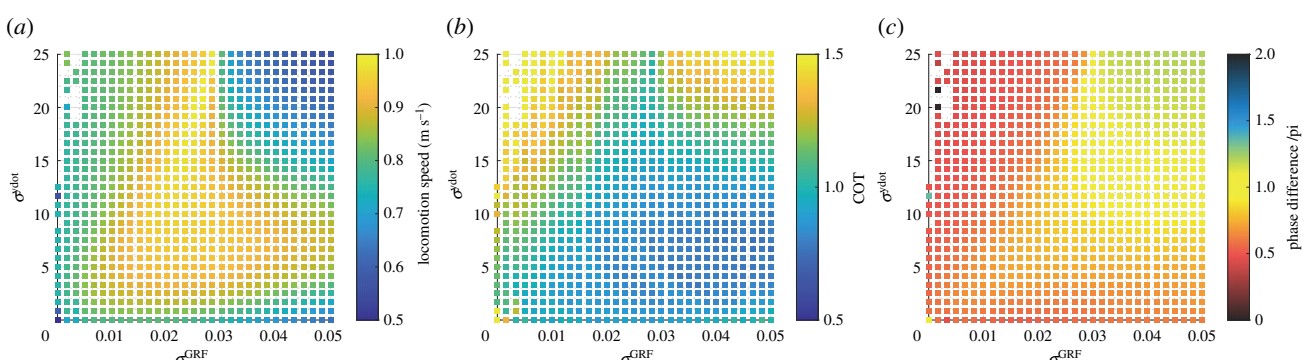

**Figure 5.** Effects of the proposed feedback mechanism on the running task. (a) Locomotion speed, (b) cost of transport and (c) phase difference between the fore and hind limbs, $\Delta\phi$. In all figures, the x- and y-axes represent the values of feedback gain for the GRF and velocity feedback, $\sigma^{\text{GRF}}$ and $\sigma^{\dot{y}}$, respectively. The other control parameters are the same as those in the experiment shown in figure 4 (i.e. table S1). The robot achieves fast and efficient running with a combination of gains around ($\sigma^{\text{GRF}}$, $\sigma^{\dot{y}}$) = (0.025, 5.0). (Online version in colour.)

phase modulation in the swing phase. In the experimental setup, after the robot ran for a specific period, the ground dropped to a height of $H^{\text{gap}}$, as shown in figure 6a. Specifically, when the simulation time exceeded a specific time $T_{\text{drop}}$ and the robot was in the flying phase (i.e. no ground contact), the

ground drop randomly occurred for each trial. We evaluated the success rate based on whether the robot continued to run after landing on the dropped ground. The parameters were the same as those in the previous running experiment (electronic supplementary material, table S1).

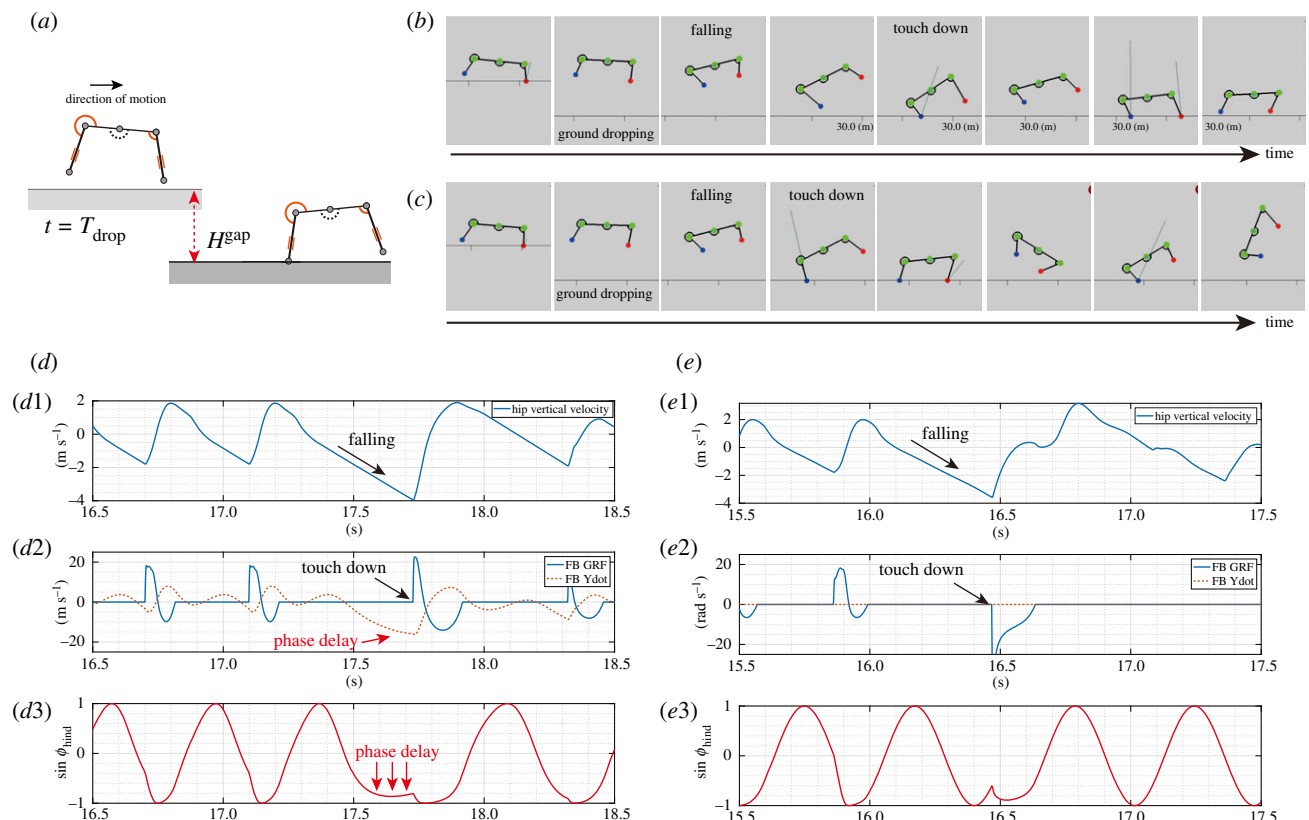

**Figure 6.** Comparison of the proposed model in the falling experiment. (*a*) Experimental setup for a falling task. While the robot is running, the level of the ground drops to a specific height $H^{\text{gap}}$. The ground drop randomly occurs when the robot is in the flying phase after the simulation time exceeds $T_{\text{drop}}$. Snapshots of the falling robot (*b*) with the GRF and velocity sensory feedback mechanism $((\sigma^{\text{GRF}}, \sigma^{\dot{y}}) = (0.025, 5.0))$ and with only the GRF feedback mechanism $((\sigma^{\text{GRF}}, \sigma^{\dot{y}}) = (0.025, 0.0))$. While the robot with the velocity sensory feedback continues to run (*b*), the robot without the velocity feedback fails to touch down, resulting in a flopped posture (*c*). The profiles show the effects of the sensory feedback mechanism during the falling task with (*d*1−3) and without (*e*1−3) the velocity sensory feedback. (*d*1) The profile shows the changes in the vertical velocity of the hip point mass. (*d*2) presents the phase modulations for the hind limb oscillator by the GRF (rigid line) and velocity (break line). (*d*3) The phase of the hind limb oscillator was delayed during falling, and the hind limb around $\phi_{\text{hind}} = 3\pi/2$. (*e*1) The profile of the vertical velocity of the hip mass point indicates that the robot falls as in (*d*1). (*e*2) While falling, there is no sensory modulation of the hind limb oscillator. (*e*3) The phase of the hind limb oscillator advances while falling, and the robot touches down when the hind limb phase $\phi_{\text{hind}}$ becomes closer to $2\pi$ than that in (*d*3). (Online version in colour.)

The distinct behaviours with and without the velocity feedback at the moment of the robot landing are shown in figures 6*b*,*c*. The robot with the velocity feedback modulates the phase of the limb, as shown in figure 6*d*2; thus, the robot touches the ground with the limb posture in the anterior position. Consequently, the robot continued to run after the falling test, achieving a spontaneous transition between steady running and the preliminary posture for landing. By contrast, the robot without the velocity feedback processes the limb phase while falling, as shown in figure 6*e*2, and flops down after touchdown. These results suggest that the proposed model allows the robot to smoothly translate from running to landing depending on the magnitude of the vertical velocity of the body part.

In addition, we evaluated the effect of the feedback terms on the falling task with various combinations of feedback gaits, $\sigma^{\text{GRF}}$ and $\sigma^{\dot{y}}$ for the altered gap heights $H^{\text{drop}}$. Figure 7 presents the success ratio, where each gain combination has ten trials. At low drop heights (e.g. $H^{\text{gap}} = 0.1$), both robots with and without the velocity feedback can adapt to the ground drop, as shown in figure 7*a*. By contrast, when $H^{\text{gap}} = 0.7$, the robot with the velocity feedback maintains a high success ratio, whereas the success ratio of the robot with only the GRF feedback is reduced (figure 7*d*). Therefore, these results suggest that the sensory feedback

based on the vertical velocity helps the robot to land the unintended fall during the running gait.

## (c) Adaptation to body morphology

Quadrupeds exhibit large variations in gait patterns, which are reasonable for their morphologies. To evaluate the adaptability of the proposed model to different morphologies, we conducted a robot running experiment with an altered body aspect ratio (i.e. the ratio of lengths between the trunk and limb, $R = L^{\text{trunk}}/L^{\text{limb}}$). Specifically, by setting the body lengths to 0.8, 1.0 and 1.4, we investigated the relationship between the morphology of the robot and the emerging running patterns. Regarding the other parameters, we employed the same parameter set shown in electronic supplementary material, table S1.

The results of the grid search indicate that the proposed model allows the robot to generate various running patterns in response to its limb length and body length, as shown in figure 8*a*. For example, when the robot has a high aspect ratio (i.e. $R = 2.8$), the robot tends to exhibit a cheetah-like running gait where there are two distinct flying phases after the lift-off of both the fore and hind limbs (figure 8*b*1 and *b*2). Furthermore, when the robot has an intermediate value of the aspect ratio (i.e. $R = 1.9$), the robot tends to

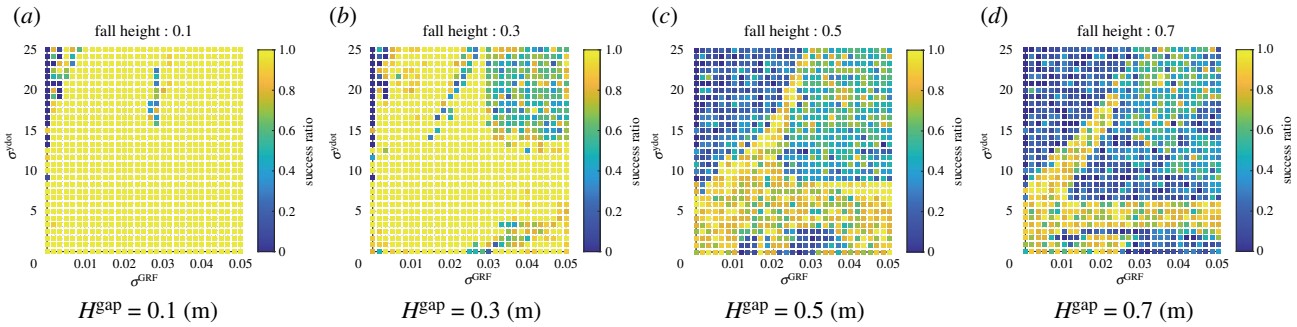

**Figure 7.** Effects of the proposed feedback mechanism on the success ratio in the falling task with altered gaps in the dropping height $H^{\text{gap}}$ = (a) 0.1, (b) 0.3, (c) 0.5 and (d) 0.7 (m). The parameters were set as listed in table S1. For each combination of the feedback gains, we conducted 10 trials. (Online version in colour.)

**Figure 8.** Various typical running patterns depending on body shapes. (a) Phase difference between the fore and hind limbs, $\Delta\phi$, with different limb and body lengths. The other parameters are set as listed in table S1. The white blank in (a) indicates that the robot failed to run. (b1) The gait diagram and (b2) the sequence of cheetah-like running gait by the robot with a high body aspect ratio ($R = 2.8$). (c1) The gait diagram and (c2) the sequence of horse-like running gait by the robot with an intermediate body aspect ratio ($R = 1.9$). (d1) The gait diagram and (d2) the sequence of pronking gait by the robot with a low body aspect ratio ($R = 1.6$). In all typical gaits (b–c), the combination of the feedback gains is the same $(\sigma^{\text{GRF}}, \sigma^{\dot{y}}) = (0.025, 5.0)$. In all gait diagrams, the coloured region represents that the foot is on the ground ($N_i^{\text{v}} > 0$). (Online version in colour.)

exhibit a horse-like running gait where there is a primal flying phase after the lift-off of the forelimb (figure 8c1 and c2). In addition, as shown in the results of the aforementioned running experiment, when the robot has a low aspect ratio (e.g. $R = 1.6$), the robot tends to exhibit a pronking gait where the fore and hind limbs move nearly synchronously (figure 8d1 and d2). Consequently, the proposed decentralized control mechanism can generate various running patterns in response to the robot morphology.

## 4. Discussion

The simulation results for the steady running and adaptation to the ground drop indicate that the proposed sensory feedback

model works for different functions regarding the efficiency or adaptability, depending on the situation. During steady locomotion, the velocity feedback mechanism adjusts the phase of the limb motion in response to the body part (e.g. the shoulder or hip). This sensory modulation allows the robot to exploit the flying phase, where the robot has no ground contact, for a faster and more efficient locomotion (figure 4c) than running by only the sensory feedback based on GRF. By contrast, during non-steady locomotion, such as falling, the large negative value of the vertical velocity strongly modulates the phase of the oscillator, as shown in figure 3b; thus, the robot prepares for landing with the limb positioning in the anterior posture. These results suggest that steady and non-steady running motions may share a simple control mechanism that exhibits situation-dependent functionalities.

The proposed model captures the trend between the morphology and running patterns in quadruped running [1,24]. For example, the pronking springbok bends its spine, and its morphology qualitatively corresponds to a robot with a low aspect ratio, as shown in figure 8a. The horse morphology with a relatively long limb length exhibits a horse-like gallop where the hind limb initiates multiple limb stances to redirect the body motion from downwards to upwards, as shown in figure 8c2. Furthermore, the cheetah with a relatively long trunk exhibits a distinct flying phase twice during one stride cycle, as shown in figure 8b2. These agreements between the animal and robot trends in the running patterns and morphology suggest that the robot can exploit the nature of the body dynamics to generate feasible running patterns through sensory feedback mechanisms based on both the GRF and vertical motion of the body parts.

The presented modelling study provides a new possibility for a sensory-motor system in adaptive quadruped locomotion. Although the proposed model exploits the vertical velocities of the shoulder and hip regions to modulate fore and hind limb motion, respectively, there are no sensory organs that directly perceive the vertical velocities of each body part [25]. Therefore, it is natural to suppose that the proposed decentralized control mechanism can be achieved by integrating global sensory information, such as visual flow [17–19], vestibular centre and proprioceptive sensibility (figure 9). For example, the local vertical velocities can be interpreted with sensory information as follows:

$$\dot{y}_{fore} = V_{opt}^{y} \tag{4.1}$$

and

$$\dot{y}_{hind} = V_{opt}^{y} - L\dot{\theta}_{trunk}\cos\theta_{trunk}, \tag{4.2}$$

where $V^{opt}$ is the vertical velocity obtained from the visual flow, $L$ is the body length and $\theta$ is the body angle along the pitch axis. Note that the proposed study suggests a new possible sensory-motor integration involving the body tilt (e.g. $\theta_{trunk}$ and $\dot{\theta}_{trunk}$) and vertical velocity from the optical queue $V_{opt}^{y}$, while the previous bioinspired robot studies considered the body tilt information [9,12,13]. Although the integration of the multimodal sensory information for limb adjustment by quadrupeds in the proposed manner remains unclear, a recent biological study demonstrates that the hindlimb posture and vestibular information are integrated for postural stability in the rolling motion [26]. We expect that higher nervous systems (e.g. vestibular nuclei) may contribute to adjusting

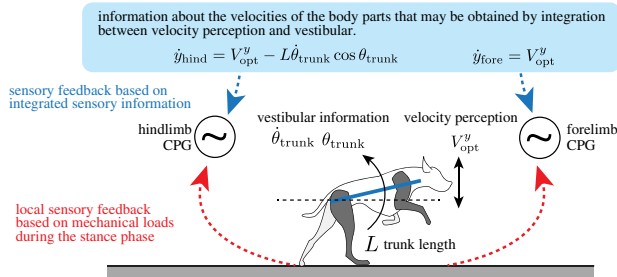

**Figure 9.** Possibility of sensory-motor structure underlying adaptive quadruped running. The local mechanical load during the stance phase (e.g. GRF) modulates each phase of the limb CPG. By contrast, the sensory information of the body part velocity may be obtained by integrating the rotational and linear motions of the entire body and modulating all CPGs in a simple manner during the swing phase. (Online version in colour.)

the limb motion in adaptive quadruped running in a manner similar to that of gaze and posture stabilization [27].

To further understand the limb control mechanism underlying adaptive quadruped running, we must consider intralimb coordination for the swing and stance phases. Regarding the modulation of the swing phase, actual quadrupeds modulate the touchdown timing and limb posture at the landing moment. For example, as the locomotion speed increases, quadrupeds exhibit a large horizontal excursion in the limb stride cycle for feasible limb properties, such as the angle of attack (AOA) and limb retraction speed [20]. We expect our proposed decentralized control mechanism to extend to the intralimb coordination mechanism by exploiting the sense of the body velocity. This is because the control model from engineering studies [28,29] demonstrates that the horizontal velocity of the centre of mass is useful for generating a reasonable AOA for the target locomotion speed.

Furthermore, intralimb coordination during the stance phase is required to comprehensively understand a quadruped's versatile locomotion involving leaping and landing. For leaping behaviour, quadrupeds need to generate large impulses by kicking the ground more strongly than in the steady running stride motion. In addition, each limb requires to change its properties for increased damping to absorb the shock from the ground while landing [30,31]. Owing to the lack of an intralimb coordination mechanism, the robot rebounded after the landing event in the failed cases of the falling task in our simulation. According to biological experiments, a falling cat activates its limb muscle more strongly prior to a landing event from higher elevations. If we can extract a simple intralimb coordination mechanism that works in both steady and non-steady behaviours depending on the situation, such as our interlimb coordination, a new control principle that allows legged robots to perform dynamic locomotion [32] without a precise model may be established.

## 5. Conclusion

To understand the essential control mechanism underlying adaptive quadruped running, this study proposed a decentralized control mechanism involving a sensory feedback mechanism during the stance and swing phases. The agreement of versatile and adaptive running by the robot in the simulation and the actual quadruped suggests that steady running and non-steady behaviours (e.g. landing)

share a common control mechanism that works differently depending on the vertical velocities of the body parts.

In a future study, we will develop a physical robot to verify the proposed model in the real world. Furthermore, we will extend the model involving intralimb coordination to coordination with body parts such as the head, tail and flexible spine, to realize further versatile and adaptive quadruped robots.

**Data accessibility.** Video files: video clips 1, 2 and 3 showing the simulation of the running experiments, falling experiments and adaptation to body morphology, respectively, are available in the electronic supplementary material [33].

**Authors' contributions.** A.F. and Y.K. conceived the experiment(s), A.F., Y.K., S.S., T.K. and A.I. developed the mathematical model, A.F., Y.K. and T.B. conducted the experiment(s), A.F., Y.K. and T.B. analysed the results. All authors reviewed the manuscript. A.F.: conceptualization, investigation, methodology, project administration, software, supervision, validation, visualization, writing-original draft, writing-review & editing; Y.K.: investigation, methodology, software; T.B.: investigation, methodology, software; S.S.: methodology, writing-review & editing; T.K.: methodology, supervision, writing-review & editing; A.I.: funding acquisition, methodology, writing-review & editing.

**Competing interests.** We declare we have no competing interests.

**Funding.** This study was supported by a JSPS KAKENHI Grant-in-Aid for Scientific Research on Grant-in-Aid for Challenging Research Exploratory under grant no. 19K21974.

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
