## [Peer Review File · Proceedings of the Royal Society B: Biological Sciences]

Review History

RSPB-2020-2822.R0 (Original submission)

Review form: Reviewer 1

Recommendation

Accept with minor revision (please list in comments)

Scientific importance: Is the manuscript an original and important contribution to its field?

Good

General interest: Is the paper of sufficient general interest?

Acceptable

Quality of the paper: Is the overall quality of the paper suitable?

Good

Is the length of the paper justified?

Yes

Should the paper be seen by a specialist statistical reviewer?

No

Do you have any concerns about statistical analyses in this paper? If so, please specify them explicitly in your report.

No

It is a condition of publication that authors make their supporting data, code and materials available - either as supplementary material or hosted in an external repository. Please rate, if applicable, the supporting data on the following criteria.

Is it accessible?

N/A

Is it clear?

N/A

Is it adequate?

N/A

Do you have any ethical concerns with this paper?

No

Comments to the Author

The authors introduce a new, heuristically defined pattern generator coupling term for the interlimb coordination in quadrupedal running, associated with the instantaneous vertical velocity of the shoulder/hip during the flight phase. The shoulder velocity term is added on top of a coupling term related to the ground reaction forces during the stance phase. They test the new interlimb coordination scheme compared to a previously published controller that relies on stance-phase feedback only (Fig 4, one experiment ?). The figure shows that the newly introduced coupling term increases the model's speed by increasing the fore-aft phase shift.

All experiments are conducted in a computer simulation, with a quadruped model simplified to its central plane.

The controller is then tested in three scenarios; running on flat ground, running and dropping at a step-down, and morphologies with changing hip-shoulder lengths; for all three scenarios varying gains are tested in a grid search. Changing gains lead to big changes in the observed fore-aft phase and success rate when dropping.

Feedback for the authors:

You propose an exciting and effective coupling term, which seemingly leads to higher robustness and quadrupedal running efficiency. Your reasoning for establishing a flight-phase coupling term is well-founded.

The number of experiments behind Fig 5, 8, and 9 (10 per data point) seems sufficient. All figures are well made and easy to understand. Most parameters are provided.

Questions to the authors:

- Equation 2, F_{trunk} , are you defining it? I could not find it.
- Equation 10 (COT); D is here a distance; in previous equations, you used D as damping constant. To avoid confusion, you could use different letters.
- Equation 12; what is z ?
- The conclusion states: '... works differently depending on the magnitude of sensory information about the vertical motion of the body.' More precisely, you are relying on velocities. Was it your intention to provide a less accurate description of your method here in the conclusion?

Literature:

The literature cited is adequate, though very short. Prof. Kimura's team has published pattern generator feedback in quadrupeds and step-downs and should be considered.

Requested changes:

- The experiment of Fig4; how many times was the experiment repeated, i.e., with a change of its initial conditions? In case it was only conducted 1x, please provide a measure of its success with changing initial conditions (i.e., 10x).
- Fig 9; the x-labels of a) and c) are not correct
- The manuscript and also the videos contain several typographic errors, and would need correction.

Videos:

All three videos are helpful and well made.

Review form: Reviewer 2

Recommendation

Major revision is needed (please make suggestions in comments)

Scientific importance: Is the manuscript an original and important contribution to its field?

Good

General interest: Is the paper of sufficient general interest?

Good

Quality of the paper: Is the overall quality of the paper suitable?

Good

Is the length of the paper justified?

No

Should the paper be seen by a specialist statistical reviewer?

No

Do you have any concerns about statistical analyses in this paper? If so, please specify them explicitly in your report.

No

It is a condition of publication that authors make their supporting data, code and materials available - either as supplementary material or hosted in an external repository. Please rate, if applicable, the supporting data on the following criteria.

Is it accessible?

N/A

Is it clear?

N/A

Is it adequate?

N/A

Do you have any ethical concerns with this paper?

No

Comments to the Author

I have now considered the contribution by Fukuhara et al. on 'Simple Decentralized Coordination Mechanism that enables limb adjustment for adaptive quadrupedal running', submitted for the special issue of PRSB on 'Stability and manoeuvrability in animal movement: lessons from biology, modelling and robotics'. I read this manuscript with this 'special issue' in mind, yet still from the perspective that the Proceedings aim at the broad biological audience (I'm part of). I also prefer that papers can be read on their own without being obliged to study earlier publications first. Although, I do understand the general message (independent feedback during stance and swing phase at the pectoral and pelvic limbs can provide stable bounding), I (being a biologist interested in locomotor functional morphology and behavior) found it difficult to grasp all details and to clearly see the 'lessons' from the modelling/robotics for biology (except from the fact that decentralized control adds to whole body performance and stability). This is of course a personal view, but I believe the authors will address a much broader readership if these general considerations can be taken into account.

Another general consideration is that I was misled by the title. I don't see the (decentralized) interlimb coordination mechanism. My interpretation is that there is simple decentralized control (local feedback) of limb movements (oscillation) that leads, via intrinsic body dynamics, to patterns of interlimb coordination. For me, this is not the same as a 'interlimb coordination mechanism'. This is also reflected in the fact that equation 9 is separate for the fore- and hind limb and that there is no mathematical formulation (mechanism) linking both to each other. This said, however, it is of course very interesting to learn that local feedback can lead to coordinated locomotion!

The more specific questions primarily concern the problems I encountered to entirely understand the model, hence to frame the simulation results. Consequently, it asked a lot of efforts and time to unravel (at least, I hope I succeeded in this) the message of the contribution. According to me, these confusing aspects must be elucidated in order to reach all potentially interested readers.

- About the trunk joint. It is said that the spine is actuated with a combination of rotatory spring and damper. According to me, however, these are passive elements (i.e. no 'motor'-function = actuation). Moreover, it is mentioned that stiffness and damping coefficient are 'set to be large' to keep the joint 'rigid'. If so, why is there a joint anyway?
- In the text the overbar above L_{trunk} (as natural length) is missing. The use of the overbar notation is a confusing. When used for the limbs, for instance, overbar L (seems to) represent(s) the variable limb length (called then the target length).
- About limb actuation and initiation of gait. It is said that the limb is actuated by changing its natural angle and length (first lines of 'Foot trajectory' paragraph). This is confusing. I interpret 'change of natural angle and length' as if the torque and linear spring are kept unloaded throughout the cycle. This does not accord to what is mentioned in equations 3 and 4 (and 5 and 6). Presumably, the first lines should read '...changing the target angle and length ...'?
- How are simulations actually initiated (i.e. for instance, period 0-1.5 s in figure 4 a,b)? Just switching on the oscillators (at the required target frequency) with the limbs in square stance? Are there other initial conditions set? Are oscillators gradually speeding up?
- About the Interlimb coordination mechanism. I'm confused by the mixed of dot-notation and separate variable names for angular speeds. My interpretation is that Φ -dot is used for the effective (target) angular speed of the oscillator and ω for, what is called, the intrinsic angular velocity (the latter without being further defined). I assume that intrinsic angular velocity represents kind of the intended speed of the oscillator (as set by the intended frequency of the

oscillator)? Adding to the confusion is that, in the figures 2 and 3, ω was drawn as a linear velocity vector.

- It is a bit strange that the 10-12 are not part of the methods section.
- In the second line of the second paragraph of the section on Interlimb coordination, I assume it should be: 'Thus, we do not explore this point...' instead of 'Thus, we do explore this point...'

From the biological side, I have one major question. It is mentioned (model description) that there is no proof that real quadrupeds have sensors detecting the (vertical) velocities of the hip and shoulder which are the control variables for the feedback. Further (see also figure 10) it is suggested that visual and vestibular information may provide this. This may be intuitive for the front parts of the body, but less evident for the pelvis and hind limbs. It could be nice if the authors could elaborate a bit more on the biological relevance of the robotic model in the discussion. Can they think of alternative (derivatives of) physiologically assessable variables that might work similarly as vertical speed?

Concerning the last simulations (changing ratio of limb and trunk length) and especially when watching the demo-videos, I found it difficult to recognize the different 'gaits' mentioned (pronking, horse-like and cheetah-like). Clearly, the interlimb coordination do differ (in terms $\Delta\Phi$) but I don't see the 'pronk' in the video (I expect 'target' angles of fore and hind identical throughout the cycle); neither can I observe two flight phase in the 'cheetah-like' gait. It could be helpful to describe better on what basis the gait-labels are given. (Notice that in panel c of figure 9 the label of the x-axis must be feedback gain (based on GRF) and not GRF).

Decision letter (RSPB-2020-2822.R0)

15-Jan-2021

Dear Dr Fukuhara:

I am writing to inform you that your manuscript RSPB-2020-2822 entitled "Simple Decentralized Interlimb Coordination Mechanism That Enables Limb Adjustment for Adaptive Quadruped Running" has, in its current form, been rejected for publication in Proceedings B.

This action has been taken on the advice of referees, who have recommended that substantial revisions are necessary. With this in mind we would be happy to consider a resubmission, provided the comments of the referees are fully addressed. However please note that this is not a provisional acceptance.

- 1) A 'response to referees' document including details of how you have responded to the comments, and the adjustments you have made.

- 2) A clean copy of the manuscript and one with 'tracked changes' indicating your 'response to referees' comments document.
- 3) Line numbers in your main document.
- 4) Data - please see our policies on data sharing to ensure that you are complying (<https://royalsociety.org/journals/authors/author-guidelines/#data>).

Sincerely,
 Dr Locke Rowe
 mailto: proceedingsb@royalsociety.org

Associate Editor

Comments to Author:

Thank you for submitting your paper for consideration for our theme issue on stability and maneuverability in animal movement. Your paper has been reviewed by two experts who are positive of the overall scientific contribution of the work for understanding the potential roles of local feedback mechanisms for generating coordinated and adaptive quadrupedal movement. However, they raise the important concern that the paper is not written in a way that will effectively convey the important details and take-home points to a broad biological audience. The specific points raised mostly concern the writing clarity and mathematical notation, and are unlikely to require completely new analysis. However, because the comments do require substantial revision to clarify details and conceptual points, I will need to send the manuscript back to review. I hope you will be able to resubmit a revised version that thoroughly addresses the feedback from the Reviewers, particularly the thoughtful and specific suggestions of Reviewer 2. Please also provide a point-by-point response to the reviewers' comments with your resubmitted paper.

Reviewer(s)' Comments to Author:

Referee: 1

Comments to the Author(s)

The authors introduce a new, heuristically defined pattern generator coupling term for the interlimb coordination in quadrupedal running, associated with the instantaneous vertical velocity of the shoulder/hip during the flight phase. The shoulder velocity term is added on top of a coupling term related to the ground reaction forces during the stance phase. They test the new interlimb coordination scheme compared to a previously published controller that relies on stance-phase feedback only (Fig 4, one experiment ?). The figure shows that the newly introduced coupling term increases the model's speed by increasing the fore-aft phase shift.

All experiments are conducted in a computer simulation, with a quadruped model simplified to its central plane.

The controller is then tested in three scenarios; running on flat ground, running and dropping at a step-down, and morphologies with changing hip-shoulder lengths; for all three scenarios varying gains are tested in a grid search. Changing gains lead to big changes in the observed fore-aft phase and success rate when dropping.

Feedback for the authors:

You propose an exciting and effective coupling term, which seemingly leads to higher robustness and quadrupedal running efficiency. Your reasoning for establishing a flight-phase coupling term is well-founded.

The number of experiments behind Fig 5, 8, and 9 (10 per data point) seems sufficient. All figures are well made and easy to understand. Most parameters are provided.

Questions to the authors:

- Equation 2, F_{trunk} , are you defining it? I could not find it.
- Equation 10 (COT); D is here a distance; in previous equations, you used D as damping constant. To avoid confusion, you could use different letters.
- Equation 12; what is z ?
- The conclusion states: '... works differently depending on the magnitude of sensory information about the vertical motion of the body.' More precisely, you are relying on velocities. Was it your intention to provide a less accurate description of your method here in the conclusion?

Literature:

The literature cited is adequate, though very short. Prof. Kimura's team has published pattern generator feedback in quadrupeds and step-downs and should be considered.

Requested changes:

- The experiment of Fig4; how many times was the experiment repeated, i.e., with a change of its initial conditions? In case it was only conducted 1x, please provide a measure of its success with changing initial conditions (i.e., 10x).
- Fig 9; the x-labels of a) and c) are not correct
- The manuscript and also the videos contain several typographic errors, and would need correction.

Videos:

All three videos are helpful and well made.

Referee: 2

Comments to the Author(s)

I have now considered the contribution by Fukuhara et al. on 'Simple Decentralized Coordination Mechanism that enables limb adjustment for adaptive quadrupedal running', submitted for the special issue of PRSB on 'Stability and manoeuvrability in animal movement: lessons from biology, modelling and robotics'. I read this manuscript with this 'special issue' in mind, yet still from the perspective that the Proceedings aim at the broad biological audience (I'm part of). I also prefer that papers can be read on their own without being obliged to study earlier publications first. Although, I do understand the general message (independent feedback during stance and swing phase at the pectoral and pelvic limbs can provide stable bounding), I (being a biologist interested in locomotor functional morphology and behavior) found it difficult to grasp all details and to clearly see the 'lessons' from the modelling/robotics for biology (except from the fact that decentralized control adds to whole body performance and stability). This is of course a personal view, but I believe the authors will address a much broader readership if these general considerations can be taken into account.

Another general consideration is that I was misled by the title. I don't see the (decentralized) interlimb coordination mechanism. My interpretation is that there is simple decentralized control (local feedback) of limb movements (oscillation) that leads, via intrinsic body dynamics, to patterns of interlimb coordination. For me, this is not the same as a 'interlimb coordination mechanism'. This is also reflected in the fact that equation 9 is separate for the fore- and hind limb and that there is no mathematical formulation (mechanism) linking both to each other. This said, however, it is of course very interesting to learn that local feedback can lead to coordinated locomotion!

The more specific questions primarily concern the problems I encountered to entirely understand the model, hence to frame the simulation results. Consequently, it asked a lot of efforts and time

to unravel (at least, I hope I succeeded in this) the message of the contribution. According to me, these confusing aspects must be elucidated in order to reach all potentially interested readers.

- About the trunk joint. It is said that the spine is actuated with a combination of rotatory spring and damper. According to me, however, these are passive elements (i.e. no 'motor'-function = actuation). Moreover, it is mentioned that stiffness and damping coefficient are 'set to be large' to keep the joint 'rigid'. If so, why is there a joint anyway?

- In the text the overbar above L_{trunk} (as natural length) is missing. The use of the overbar notation is confusing. When used for the limbs, for instance, overbar L (seems to) represent(s) the variable limb length (called then the target length).

- About limb actuation and initiation of gait. It is said that the limb is actuated by changing its natural angle and length (first lines of 'Foot trajectory' paragraph). This is confusing. I interpret 'change of natural angle and length' as if the torque and linear spring are kept unloaded throughout the cycle. This does not accord to what is mentioned in equations 3 and 4 (and 5 and 6). Presumably, the first lines should read '...changing the target angle and length ...'?

- How are simulations actually initiated (i.e. for instance, period 0-1.5 s in figure 4 a,b)? Just switching on the oscillators (at the required target frequency) with the limbs in square stance? Are there other initial conditions set? Are oscillators gradually speeding up?

- About the Interlimb coordination mechanism. I'm confused by the mixed of dot-notation and separate variable names for angular speeds. My interpretation is that $\Phi\dot{}$ is used for the effective (target) angular speed of the oscillator and ω for, what is called, the intrinsic angular velocity (the latter without being further defined). I assume that intrinsic angular velocity represents kind of the intended speed of the oscillator (as set by the intended frequency of the oscillator)? Adding to the confusion is that, in the figures 2 and 3, ω was drawn as a linear velocity vector.

- It is a bit strange that the 10-12 are not part of the methods section.

- In the second line of the second paragraph of the section on Interlimb coordination, I assume it should be: 'Thus, we do not explore this point...' instead of 'Thus, we do explore this point...'.

From the biological side, I have one major question. It is mentioned (model description) that there is no proof that real quadrupeds have sensors detecting the (vertical) velocities of the hip and shoulder which are the control variables for the feedback. Further (see also figure 10) it is suggested that visual and vestibular information may provide this. This may be intuitive for the front parts of the body, but less evident for the pelvis and hind limbs. It could be nice if the authors could elaborate a bit more on the biological relevance of the robotic model in the discussion. Can they think of alternative (derivatives of) physiologically assessable variables that might work similarly as vertical speed?

Concerning the last simulations (changing ratio of limb and trunk length) and especially when watching the demo-videos, I found it difficult to recognize the different 'gaits' mentioned (pronking, horse-like and cheetah-like). Clearly, the interlimb coordination do differ (in terms $\Delta\Phi$) but I don't see the 'pronk' in the video (I expect 'target' angles of fore and hind identical throughout the cycle); neither can I observe two flight phase in the 'cheetah-like' gait. It could be helpful to describe better on what basis the gait-labels are given. (Notice that in panel c of figure 9 the label of the x-axis must be feedback gain (based on GRF) and not GRF).

Author's Response to Decision Letter for (RSPB-2020-2822.R0)

See Appendix A.

RSPB-2021-1622.R0

Review form: Reviewer 2

Recommendation

Accept with minor revision (please list in comments)

Scientific importance: Is the manuscript an original and important contribution to its field?

Good

General interest: Is the paper of sufficient general interest?

Good

Quality of the paper: Is the overall quality of the paper suitable?

Good

Is the length of the paper justified?

Yes

Should the paper be seen by a specialist statistical reviewer?

No

Do you have any concerns about statistical analyses in this paper? If so, please specify them explicitly in your report.

No

It is a condition of publication that authors make their supporting data, code and materials available - either as supplementary material or hosted in an external repository. Please rate, if applicable, the supporting data on the following criteria.

Is it accessible?

N/A

Is it clear?

N/A

Is it adequate?

N/A

Do you have any ethical concerns with this paper?

No

Comments to the Author

I very much appreciate the efforts made by the authors to address all my queries. I believe this paper is more accessible now for the broader audience (yet, remains very specialised).

Two small suggestion remains:

- In equation 9, I suggest to use the mathematical dot-notation also in the superscript of sigma (gain of the velocity feedback).
- Lines 73-77 remains unclear to me. I do understand the aspect of mass-distribution, but don't see why it is necessary to include a spring and damper with coefficients that are that large that the trunk behaves as one rigid element. Is this just a matter of coding or are their mechanical reasons for this. Please explain this better .

Decision letter (RSPB-2021-1622.R0)

09-Sep-2021

Dear Dr Fukuhara

I am pleased to inform you that your manuscript RSPB-2021-1622 entitled "Simple Decentralized Coordination Mechanism That Enables Limb Adjustment for Adaptive Quadruped Running" has been accepted for publication in Proceedings B.

The referee(s) have recommended publication, but also suggest some minor revisions to your manuscript. Therefore, I invite you to respond to the referee(s)' comments and revise your manuscript. Because the schedule for publication is very tight, it is a condition of publication that you submit the revised version of your manuscript within 7 days. If you do not think you will be able to meet this date please let us know.

Sincerely,

Dr Locke Rowe

Associate Editor

Board Member

Comments to Author:

Dear authors,

Thank you so much for thoroughly addressing the reviewers' feedback on the previous version of your paper. Your revised paper has received feedback from an expert reviewer. They are now satisfied that the paper is more accessible to a broad audience and suggest just a couple of minor additional clarifications to the text. I am happy to accept the paper for publication in the special

issue on Stability and Maneuverability in Animal Movement, subject to addressing these additional minor comments in the final submitted version.

Reviewer(s)' Comments to Author:

Referee: 2

Comments to the Author(s).

I very much appreciate the efforts made by the authors to address all my queries. I believe this paper is more accessible now for the broader audience (yet, remains very specialised).

Two small suggestion remains:

- In equation 9, I suggest to use the mathematical dot-notation also in the superscript of sigma (gain of the velocity feedback).

- Lines 73-77 remains unclear to me. I do understand the aspect of mass-distribution, but don't see why it is necessary to include a spring and damper with coefficients that are that large that the trunk behaves as one rigid element. Is this just a matter of coding or are their mechanical reasons for this. Please explain this better .

Author's Response to Decision Letter for (RSPB-2021-1622.R0)

See Appendix B.

Decision letter (RSPB-2021-1622.R1)

13-Oct-2021

Dear Dr Fukuhara

I am pleased to inform you that your manuscript entitled "Simple Decentralized Coordination Mechanism That Enables Limb Adjustment for Adaptive Quadruped Running" has been accepted for publication in Proceedings B.

Data Accessibility section

Open Access

Paper charges

Sincerely,

Appendix A

Manuscript ID: RSPB-2020-2822

Simple Decentralized Interlimb Coordination Mechanism That Enables Limb Adjustment for Adaptive Quadruped Running

Authors: Akira Fukuhara, Yukihiro Koizumi, Tomoyuki Baba, Shura Suzuki, Takeshi Kano, and Akio Ishiguro

We would like to thank all reviewers for your time and effort and for your useful and constructive comments to further improve the overall quality of our paper. We have carefully considered the suggestions made by the reviewers and have included corresponding modifications in the revised manuscript. Detailed replies to all reviewer comments can be found below.

Response to Reviewer 1

Comment 1-1

The authors introduce a new, heuristically defined pattern generator coupling term for the interlimb coordination in quadrupedal running, associated with the instantaneous vertical velocity of the shoulder/hip during the flight phase. The shoulder velocity term is added on top of a coupling term related to the ground reaction forces during the stance phase. They test the new interlimb coordination scheme compared to a previously published controller that relies on stance-phase feedback only (Fig 4, one experiment?). The figure shows that the newly introduced coupling term increases the model's speed by increasing the fore-aft phase shift.

All experiments are conducted in a computer simulation, with a quadruped model simplified to its central plane.

The controller is then tested in three scenarios; running on flat ground, running and dropping at a step-down, and morphologies with changing hip-shoulder lengths; for all three scenarios varying gains are tested in a grid search. Changing gains lead to big changes in the observed fore-aft phase and success rate when dropping.

Feedback for the authors:

You propose an exciting and effective coupling term, which seemingly leads to higher robustness and quadrupedal running efficiency. Your reasoning for establishing a flight phase coupling term is well-founded. The number of experiments behind Fig 5, 8, and 9 (10 per data point) seems sufficient. All figures are well made and easy to understand. Most parameters are provided.

Response to comment 1-1:

Thank you for the fruitful comments. We really appreciate the detailed review, especially the per-section feedback. We modified the text based on the comments provided.

Comment 1-2

Questions to the authors:

- Equation 2, F_{trunk} , are you defining it? I could not find it.

Equation 10 (COT); D is here a distance; in previous equations, you used D as damping constant. To avoid confusion, you could use different letters.

Equation 12; what is z ?

Response to comment 1-2:

We apologise for the lack of explanation and poor definition of the parameters. Regarding F_{trunk} , we added an explanation of the parameters for the sake of clarity (Line 77).

Regarding the definition of COT, we certainly used the misleading label D . This label was changed for the running distance from D to X . (Line 142 and Equation (10))

Regarding z , it is just an argument for the function $\chi(z)$, which returns the value of the argument z if z is positive. Otherwise, $\chi(z)$ returns 0. We added the corresponding explanation (Line 144-145).

Comment 1-3

The conclusion states: ‘... works differently depending on the magnitude of sensory information about the vertical motion of the body.’ More precisely, you are relying on velocities. Was it your intention to provide a less accurate description of your method here in the conclusion?

Response to comment 1-3:

Thank you for this comment. Regarding the sentence mentioned, it was an unintended mistake because an object has velocity when it is moving. To solve this misunderstanding, we revised the statement in the conclusion as follows:

“The agreement of the versatile and adaptive running by the robot in both the simulation and actual motion of the quadruped suggests that steady running and non-steady behaviours (e.g., landing) share a common control mechanism that works differently depending on **the vertical velocities of the body parts.**” (Lines 299-300).

Comment 1-4

Literature:

The literature cited is adequate, though very short. Prof. Kimura’s team has published pattern generator feedback in quadrupeds and step-downs and should be considered.

Response to comment 1-4:

Thank you for the kind suggestion. We added references about the bio-inspired control mechanism with modulation in the swing phase, including Kimura et al. In previous bio-inspired studies, the limb stride motion in the swing phase was modulated based on the trunk posture or collision between the foot and obstacles. The vestibular modulation, which Kimura proposed, plays an essential role in the posture control and gait transition of quadruped robots. However, there is an open question regarding the essential control mechanism required during the swing phase for steady and non-steady quadruped locomotion, e.g., leaping and landing.

In contrast to previous studies, our control mechanism can regulate the coordination of fore and hind limbs based on the vertical velocities of the shoulder and hip for adaptive quadruped running and landing. The proposed mechanism could be interpreted as a multi-sensory feedback mechanism based on rotational motion from the vestibular system (as in previous studies) and translational motion from the visual system. We believe that our proposed model suggests new insight on multi-sensory integration for adaptive quadruped running.

We added new references in the Introduction (Lines 31-33) and discussed the relationship between the proposed model and previous bio-inspired studies (Lines 275-276).

Comment 1-5

Requested changes:

- The experiment of Fig4; how many times was the experiment repeated, i.e., with a change of its initial conditions? In case it was only conducted 1x, please provide a measure of its success with changing initial conditions (i.e., 10x).

Response to comment 1-5:

Thank you for the comment. As the reviewer mentions, the previous manuscript lacked an explanation about the initial condition. The initial robot position is given by a height of 0.5 [m] so that the robot has no ground contact. Furthermore, the robot has no initial forward velocity of the body, and therefore the robot falls before the first touch-down event. Regarding the phase of the limb controller, the robot has a specific initial phase $(\phi_{\text{fore}}, \phi_{\text{hind}}) = (0.5\pi, 1.5\pi)$. The value of ω increases from 0.0 [rad/s] to a specific value (13.75 [rad/s]) at the beginning of the running simulation. We added information about the initial condition of the simulation (Lines 177-181).

Regarding Fig4., the results show a single trial with a specific initial phase $(\phi_{\text{fore}}, \phi_{\text{hind}}) = (0.5\pi, 1.5\pi)$. We also observed the same effects of the feedback term for additional evaluations with random initial phases based on the vertical velocities of body parts in ten trials. Consequently, the robot improves the average locomotion from 1.57 to 1.96 [m/s] and the average COT from 1.01 to 0.80, depending on the changes in the σ^{ydot} value from 0.0 to 5.0. We added the results of ten trials with random initial phases (Lines 189–191).

Comment 1-6

Fig 9; the x-labels of a) and c) are not correct.

Response to comment 1-6:

Thank you for the kind suggestion. We revised the labels in Fig. 9.

Comment 1-7

The manuscript and also the videos contain several typographic errors, and would need correction.

Response to comment 1-7:

Thank you for the comment. We revised the typos in the manuscript and videos. Please note the highlighted parts in the manuscript and videos.

Comment 1-8

Videos: All three videos are helpful and well made.

Response to comment 1-8:

Thank you for checking the videos. We revised the typos in the videos and replaced a part of a video to show differences in running patterns, pronking gait, horse-like, and cheetah-like running.

Response to Reviewer 2

Comment 2-1

I have now considered the contribution by Fukuhara et al. on ‘Simple Decentralized Coordination Mechanism that enables limb adjustment for adaptive quadrupedal running’, submitted for the special issue of PRSB on ‘Stability and manoeuvrability in animal movement: lessons from biology, modelling and robotics’. I read this manuscript with this ‘special issue’ in mind, yet still from the perspective that the Proceedings aim at the broad biological audience (I’m part of). I also prefer that papers can be read on their own without being obliged to study earlier publications first. Although, I do understand the general message (independent feedback during stance and swing phase at the pectoral and pelvic limbs can provide stable bounding), I (being a biologist interested in locomotor functional morphology and behavior) found it difficult to grasp all details and to clearly see the ‘lessons’ from the modelling/robotics for biology (except from the fact that decentralized control adds to whole body performance and stability). This is of course a personal view, but I believe the authors will address a much broader readership if these general considerations can be taken into account.

Response to comment 2-1:

Thank you for the suggestions. As the reviewer mentions, the previous manuscript did not clearly convey a take-home message for prospective audience in the field of biology. By exploring the decentralized control mechanism that leverages vertical velocities of the anterior and posterior body parts (e.g., shoulder and hip), we found a possibility of multi-modal sensory integration for adaptive quadrupedal running. The vertical velocities of the shoulder and hip (\dot{y}_{fore} and \dot{y}_{hind}) can be described according to the following multi-modal sensory information:

$$\dot{y}_{\text{fore}} = V_{\text{opt}}^y, \quad (1)$$

$$\dot{y}_{\text{hind}} = V_{\text{opt}}^y - L\dot{\theta}_{\text{trunk}} \cos \theta_{\text{trunk}}, \quad (2)$$

where V^{opt} is the vertical velocity obtained from the visual flow, L is the body length, and θ_{trunk} is the body angle along the pitch axis. These translations suggest that sensory information of the vertical velocity of a body part may reflect various sensory data, e.g., visual flow, body tilt angle, and angular velocity from the vestibular system. In contrast, previous bio-inspired robotics studies reported a sensory-motor mechanism based on the body tilt angle for limb adjustment in the swing phase (Kimura et al. *The Int. J. Robotics Res.*, 2015, Fukuoka et al. *Sci. reports*, 2015, Fukui et al. *Robotics Auton. Syst.* 2019). Although it is still unclear how animals integrate their multi-modal sensory information for adaptive locomotion, we hypothesise that higher nervous systems (e.g., vestibular nuclei) may contribute to adjusting limb motion in adaptive quadrupedal running in similar terms to gaze and posture stabilizations. We added further explanation about the lesson extracted from our decentralized control study for biological fields in the discussion (Lines 267-280).

Comment 2-2

Another general consideration is that I was misled by the title. I don’t see the (decentralized) interlimb coordination mechanism. My interpretation is that there is simple decentralized control (local feedback) of limb movements (oscillation) that leads, via intrinsic body dynamics, to patterns of interlimb coordination. For me, this is not the same as a ‘interlimb coordination mechanism’. This is also reflected in the fact that equation 9 is separate for the fore- and hind limb and that there is no mathematical formulation (mechanism) linking both to each other. This said, however, it is of course very interesting to learn that local feedback can lead to coordinated locomotion!

Response to comment 2-2:

Thank you for the suggestions. We described the proposed control mechanism with the terminology, interlimb coordination mechanism because decoupled CPG models (without neural connectivity between limbs) are investigated as interlimb coordination in the Bio-inspired robotics field (Aoi, S., et al. (2017). *Adaptive control strategies for interlimb coordination in legged robots: a review.* *Frontiers in neurorobotics*, 11, 39.). However, it is confusing for readers in the biological field. So, we modified the technical term from “interlimb coordination mechanism” to “decentralized control mechanism” (e.g., Lines 106-108 and 119). And, we revised the title as follow:

The previous title:

Simple Interlimb Decentralised Control Mechanism That Enables Limb Adjustment for Adaptive Quadruped

Running

The revised title:

Simple **Decentralised Control Mechanism** That Enables Limb Adjustment for Adaptive Quadruped Running

Comment 2-3

The more specific questions primarily concern the problems I encountered to entirely understand the model, hence to frame the simulation results. Consequently, it asked a lot of efforts and time to unravel (at least, I hope I succeeded in this) the message of the contribution. According to me, these confusing aspects must be elucidated in order to reach all potentially interested readers.

Response to comment 2-3:

Thank you for the helpful comment. We revised the manuscript based on this comment. Please, check the rest of responses to reviewers' comments.

Comment 2-4

About the trunk joint. It is said that the spine is actuated with a combination of rotatory spring and damper. According to me, however, these are passive elements (i.e. no 'motor'- function = actuation). Moreover, it is mentioned that stiffness and damping coefficient are 'set to be large' to keep the joint 'rigid'. If so, why is there a joint anyway?

Response to comment 2-4:

Thank you for the comment. The explanation concerning body modelling was certainly confusing. As the reviewer mentions, the spine has a passive rotational joint. To design a simple body structure, we modelled the trunk unit as rigid. However, to adjust the mass distributions of the robot body, setting the point-mass m^{spine} in the middle of the trunk is required. Owing to the mass-spring-damper system, three point-masses, $m_{\text{fore}}^{\text{base}}$, m^{spine} , $m_{\text{hind}}^{\text{base}}$ should be connected through rotary spring and damper to keep the trunk straight. We revised the description of the hardware model (Lines 73-75).

Comment 2-5

About limb actuation and initiation of gait. It is said that the limb is actuated by changing its natural angle and length (first lines of 'Foot trajectory' paragraph). This is confusing. I interpret 'change of natural angle and length' as if the torque and linear spring are kept unloaded throughout the cycle. This does not accord to what is mentioned in equations 3 and 4 (and 5 and 6). Presumably, the first lines should read '...changing the target angle and length ...'?

Response to comment 2-5:

Thank you for the helpful comment. As the reviewer mentions, the modelling of limb actuation was certainly confusing, especially the first lines in the Foot Trajectory subsection. We revised the manuscript to explain that the limb motion is generated by changing the target angle and length of the limb rotational and prismatic actuators (Lines 82-83).

Comment 2-6

How are simulations actually initiated (i.e. for instance, period 0-1.5 s in figure 4 a,b)? Just switching on the oscillators (at the required target frequency) with the limbs in square stance? Are there other initial conditions set? Are oscillators gradually speeding up?

Response to comment 2-6:

Thank you for the comment. As the reviewer mentions, the previous manuscript lacked an explanation about the initial condition. As we mentioned in the response to Comment 1-5, the initial robot position is given by a height of 0.5 [m] so that the robot has no ground contact. Furthermore, the robot has no initial forward velocity of the body, and therefore the robot falls before the first touch-down event. Regarding the phase of the limb controller, the robot has a specific initial phase ($\phi_{\text{fore}}, \phi_{\text{hind}} = (0.5\pi, 1.5\pi)$). The value of ω increases from 0.0 [rad/s] to a specific value (13.75 [rad/s]) at the beginning of the running simulation. We added information about the initial condition of the simulation (Lines 177-181).

Comment 2-7

About the Interlimb coordination mechanism. I'm confused by the mixed of dot-notation and separate variable names for angular speeds. My interpretation is that Phi-dot is used for the effective (target) angular speed of the oscillator and omega for, what is called, the intrinsic angular velocity (the latter without being further defined). I assume that intrinsic angular velocity represents kind of the intended speed of the oscillator (as set by the intended frequency of the oscillator)? Adding to the confusion is that, in the figures 2 and 3, omega was drawn as a linear velocity vector.

Response to comment 2-7:

Thank you for the comment. Equation 9 in the main manuscript describes the time evolution of ϕ_i . Changes in ϕ_i comprise the intrinsic rhythm generator and external sensory feedback mechanism. The parameter ω (i.e., the intrinsic angular velocity) corresponds to the descending signal from the brain to the CPG network to detect the basic locomotor frequency. By changing the value of ω , the robot can change the locomotor speed: low-speed walking with low ω and high-speed running with high ω .

We added explanations in the Model section to clarify the functionality of the parameter ω (Lines 122-124). We also revised Figures 2(c) and 3 regarding visualization of ω .

Comment 2-8

It is a bit strange that the 10-12 are not part of the methods section.

Response to comment 2-8:

Thank you for the comment. We moved the measurements of the robot performance from the Result section to the Method section (Line 138).

Comment 2-9

In the second line of the second paragraph of the section on Interlimb coordination, I assume it should be: ‘Thus, we do not explore this point...’ instead of ‘Thus, we do explore this point...’.

Response to comment 2-9:

Thank you for the kind comment. We revised the sentence mentioned as follows: “Thus, we attempt to understand the substantial control mechanism by describing how sensory feedback could work simply and abstractly” (Lines 115-116).

Comment 2-10

From the biological side, I have one major question. It is mentioned (model description) that there is no proof that real quadrupeds have sensors detecting the (vertical) velocities of the hip and shoulder which are the control variables for the feedback. Further (see also figure 10) it is suggested that visual and vestibular information may provide this. This may be intuitive for the front parts of the body, but less evident for the pelvis and hind limbs. It could be nice if the authors could elaborate a bit more on the biological relevance of the robotic model in the discussion. Can they think of alternative (derivatives of) physiologically assessible variables that might work similarly as vertical speed?

Response to comment 2-10:

Thank you for the suggestions. As the reviewer mentions, we did not expect that some sensory organs directly and respectively detect vertical velocities of the anterior and posterior body parts (e.g., shoulder and hip).

We expect that quadrupeds may integrate multi-modal sensory information, e.g., their velocity perception from the optical flow and vestibular system in the higher nervous system, to implement sensory-motor modulation for each body part. Although it is still unclear how quadrupeds integrate multi-modal sensory information, a recent biological study reported that the hind-limb posture and vestibular system are integrated to achieve postural control in rolling movement (not pitching) (A.A. McCall et al. "Integration of vestibular and hindlimb inputs by vestibular nucleus neurons: multisensory influences on postural control", *The Journal of Neurophysiology*, 125, 4, pp. 1095–1110, 2021).

Concerning the response to Comment 2-10 and also 2-1, we added discussion about the possible implementation of multi-modal sensory information for adaptive quadruped locomotion (Lines 267–280).

Comment 2-11

Concerning the last simulations (changing ratio of limb and trunk length) and especially when watching the demo-videos, I found it difficult to recognize the different 'gaits' mentioned (pronking, horse-like and cheetah-like). Clearly, the interlimb coordination do differ (in terms $\Delta\Phi$) but I don't see the 'pronk' in the video (I expect 'target' angles of fore and hind identical throughout the cycle); neither can I observe two flight phase in the 'cheetah-like' gait. It could be helpful to describe better on what basis the gait-labels are given. (Notice that in panel c of figure 9 the label of the x-axis must be feedback gain (based on GRF) and not GRF).

Response to comment 2-11:

Thank you for the comments. We apologise for submitting confusing video clips showing running behaviours before the synchronization of limbs. Furthermore, the results using the previous body parameter exhibited minor differences between gait patterns. We conducted an additional grid search to evaluate the relationship between body parameters and emerging gait patterns. Based on further evaluation, we revised Figure 9 and the video clips.

Moreover, we added an explanation about gait labelling. We mainly detect three gait patterns (e.g., pronking, horse-like, and cheetah-like running) based on the phase difference between the fore and hind limbs. We also checked the number of flying phases during one stride cycle. We added an explanation about gait evaluation (Lines 149–166). Besides, we added gait diagrams to show flying phases in each running pattern in Fig. 9.

Appendix B

Manuscript ID: RSPB-2020-2822

Simple Decentralised Control Mechanism That Enables Limb Adjustment for Adaptive Quadruped Running

Authors: Akira Fukuhara, Yukihiro Koizumi, Tomoyuki Baba, Shura Suzuki, Takeshi Kano, and Akio Ishiguro

We would like to thank all reviewers for your time and effort and for your useful and constructive comments. We have revised minor points based on the review's comments.

Response to Reviewer 2

Comment 2-1

Comments to the Author(s). I very much appreciate the efforts made by the authors to address all my queries. I believe this paper is more accessible now for the broader audience (yet, remains very specialised).

Two small suggestion remains:

- In equation 9, I suggest to use the mathematical dot-notation also in the superscript of sigma (gain of the velocity feedback).
- Lines 73-77 remains unclear to me. I do understand the aspect of mass-distribution, but don't see why it is necessary to include a spring and damper with coefficients that are that large that the trunk behaves as one rigid element. Is this just a matter of coding or are their mechanical reasons for this. Please explain this better .

Response to comment 2-1:

We would like yo appreciate the suggestive comments. we revised the description of gain of the velocity feedback from σ^{ydot} to $\sigma^{\dot{y}}$.

Regarding the modeling of the trunk unit, we set the high stiffness values of prismatic link in the trunk because the trunk of actual quadrupeds rarely contracts due to the stiff axial skeleton. Although some species (e.g., cheetah) exploit the flexible bending of the spine, we employ the stiff rotational spring to omit the effect of spine bending for simplicity. We added explanation (see line 77-78).